# Blade Augmentation in Nailing Proximal Femur Fractures—An Advantage despite Higher Costs?

**DOI:** 10.3390/jcm12041661

**Published:** 2023-02-19

**Authors:** Alexander Böhringer, Raffael Cintean, Alexander Eickhoff, Florian Gebhard, Konrad Schütze

**Affiliations:** Department of Trauma, Hand and Reconstructive Surgery Ulm University, Albert-Einstein-Allee 23, 89081 Ulm, Germany

**Keywords:** proximal femur and hip fracture, tip-apex distance, helical blade position, cut-out rate, cement augmentation, osteoporosis

## Abstract

Background: Proximal femoral fractures occur with increasing incidence, especially in the elderly. Commonly used implants for surgical treatment are cephalomedullary nails. To increase stability, a perforated femoral neck blade can be augmented with cement. The study investigated whether this results in a relevant clinical advantage and justifies the higher cost. Materials and methods: This is a single-center retrospective study of 620 patients with proximal femur fractures treated with cephalomedullary nailing. Between January 2016 and December 2020, 207 male and 413 female patients were surgically treated with a proximal femur nail (DePuy Synthes) using a perforated blade and cement augmentation in cases with severe osteoporosis. Primary outcome measures were the rate of cut-out, tip apex distance and the positioning of the blade in the femoral head. Secondary outcome measures were the implant costs and operating times. Results: Of the 620 femoral neck blades, 299 were augmented with cement. A total of six cut-outs were seen in the first 3 months after the operation. There were three in the cement-augmented group (CAB = cement-augmented blade) and three in the conventional group (NCAB = non-cement-augmented blade). There was a significant positive correlation between age and augmentation, with a mean difference of 11 years between the two groups (CAB 85.7 ± 7.9 vs. NCAB 75.3 ± 15.1; *p* < 0.05). There was no difference in the tip-apex distance (CAB 15.97 vs. 15.69; *p* = 0.64) or rate of optimal blade positions between the groups (CAB 81.6% vs. NCAB 83.2%; *p* = 0.341). Operation times were significantly longer in the cemented group (CAB 62.6 21.2 min vs. NCAB 54.1 7.7 min; *p* < 0.05), and the implant cost nearly doubled due to augmentation. Conclusion: When the principles of anatomic fracture reduction, optimal tip-apex distance and optimal blade position are combined with cement augmentation in cases of severe osteoporosis, a cut-out rate of less than 1% can be achieved. Nevertheless, it should be noted that augmentation remains expensive and prolongs surgery time without definite proof of mechanical superiority.

## 1. Introduction

Proximal femur fractures occur with increasing incidence, especially in the elderly population due to demographic change, osteoporosis and increased activity level with unsteady walking [1]. Prompt surgical care within 24 h is recommended to avoid serious complications and achieve better patient outcomes [2,3]. Various surgical procedures and implants have been developed, including wires, screws, plates, and intramedullary nails. Despite several treatment options, the risk of mechanical failures, such as the femoral neck blade cutting through the femoral head, has been reported in the literature to be as high as 17% [4,5,6,7,8]. In comparison, a recent study by Bojan et al. [9] showed an osteosynthesis failure rate of only 2% using the “gamma nail” and state-of-the-art implantation techniques. Although the risk of mechanical osteosynthesis failure has been reduced, there is still a need for improvement, as the mortality rate in the first year is as high as 56% when revision surgery is required [10,11].

One of the most commonly used implants today for treating proximal femoral fractures is the PFNA (“proximal femur nail anti-rotation”; DePuy Synthes). To increase stability in the femoral head, the perforated helical femoral neck blade can be augmented with cement [12]. Biomechanical experiments have shown higher osteosynthesis stability by cement augmentation of the femoral neck blade. More test cycles to mechanical osteosynthesis failure, higher rotational stability, and higher pullout strength were achieved on the cadaveric model [13,14,15]. The first promising clinical results of cement-augmented blades (CAB) in the femoral head were shown by Kammerlander et al. [16].

In this study, no mechanical failure was observed in 105 patients with CAB. In contrast, 6 of the 118 patients in the non-cement-augmented blade (NCAB) group showed osteosynthesis failure. In a study with only 67 patients [17], there was no significant difference in complications but also fewer mechanical failures in the CAB group (CAB cut-out 2% vs. NCAB 14%). Schuetze et al. [18] demonstrated in 152 cases that PFNA blade augmentation is a safe and good procedure with no increase in complications and mortality. There was no cement leakage into the joint, while blood pressure changes occurred during augmentation. The aim of the study was to evaluate if there is an advantage and at what cost with the use of CAB compared to NCAB in a large cohort of patients.

## 2. Materials and Methods

Approval was obtained from our ethics committee for the use of patient data prior to the evaluation. All patients between January 2016 and December 2020 who had undergone PFNA surgery for proximal femur fracture were retrospectively analyzed for this study. We included all patients with radiographic follow-ups after at least 3 months. We excluded patients with pathologic fractures or additional surgeries for other injuries in the same hospital stay. For all cases, a perforated helical femoral neck blade was used (Fa. DePuy Synthes). The decision for or against cement augmentation (TRAUMACEM V+ Injectable Bone Cement system; Fa. DePuy Synthes) was made by the surgeon, taking into account patient age, fracture pattern, and intraoperative assessment of bone quality. Specific criteria included previously diagnosed and treated osteoporosis in the patient record, unstable fracture patterns, such as multiple fragments, severe dislocation, and varus tilt, as well as intraoperatively tested bone quality during guide-wire insertion and femoral neck blade reaming. Before blade augmentation with cement, a contrast agent was injected to avoid possible leakage into the hip joint. The treating surgeon decided intraoperatively on the final amount of cement (about 4–7 mL).

To investigate the post-operative outcome, radiographs were evaluated before discharge, after six weeks and after 3 months. A total of nine different blade positions were defined across two radiographic planes (anterior/posterior = a.p. and Lauenstein = axial). In a.p. view, the inferior, center or superior blade positions were evaluated. The axial view was used to determine the center, anterior or posterior. The blade position was determined, and the distance of the blade tip to the apex of the femoral head was measured as described by Baumgaertner et al. [19]. The measurement method for determining the tip-apex distance (TAD) and the blade position is illustrated by two examples in Figure 1. Radiographs at 6 weeks and 3 months were analyzed by two independent investigators for signs of mechanical failure. Patient charts, patient records, and anesthesia protocols were evaluated regarding surgical and non-surgical complications during hospital stays. Costs for cement augmentation were measured by the price of the augmentation kit (Traumacem V+ cement, Fa. DePuy Synthes) and the additional operating time. Data collection was performed with SAP using ICD-10 and OPS codes. Data analysis was performed with Microsoft Excel (2019 MSO) and IBM SPSS Statistics (V27.0). Demographic characteristics were described as means and as standard deviations. Due to the low number of mechanical failures, logistic regression was not possible. Therefore, group comparisons were performed using the chi-square test for frequencies, Fischer’s exact test for small sample sizes, and the Student’s *t*-test or Welch’s test for the comparisons of means.

## 3. Results

### 3.1. Patient Population

The medical records of the 620 patients were retrospectively reviewed. In 299 (48.2%) cases, cement augmentation of the blade was performed and was not performed in 321. The mean age was 80 ± 13 years and showed a significant difference between the groups (CAB 86 ± 7; NCAB 75 ± 15; *p* < 0.05). Out of the 620 patients, 207 were male, and 413 were female. Significantly more women were treated with cement augmentation of the blade (*p* < 0.05). In the CAB group, 78.3% of the patients were female, and 21.7% were male; in the NCAB group, 55.8% of the patients were female and 44.2% male. There was no significant difference between the two groups in terms of fracture classification according to AO (Arbeitsgemeinschaft für Osteosynthesefragen). Type 31-A2 was the most common, with 150 cases in the CAB group and 143 cases in the NCAB group. The ASA score (American Society of Anesthesiologists Classification) also showed no significant differences in the distribution of grades 1–4 in the two groups. As expected, there was no difference in time to surgery. The increased time required for cement augmentation of the femoral neck blade was significant at 8 min (*p* < 0.001). The hospital stay of both groups of patients did not differ significantly. The optimal blade position was achieved with center-center or center-inferior in both groups with >80%. Likewise, the tip-apex distance showed no significant difference in both groups, with 16 ± 7.3 mm in CAB and 15.7 ± 7.6 mm in NCAB. In each of the two groups, three cases with a blade cut-out occurred. Regarding the risks of post-operative hematoma and wound infection, there was no significant difference between the two groups (although close at *p* = 0.101 for hematoma). Finally, in hospital, death and one-year mortality rates were significantly higher in the CAB group. Follow-up was possible for 3 months in all patients and up to 1 year postoperatively in 313 patients. The main results are shown in Table 1 and Figure 2.

### 3.2. Mechanical Failure, Blade Position and Cement Augmentation

At the follow-up at 3 months, only six patients showed signs of mechanical failure. Of these, three were in the CAB group and three in the NCAB group. The respective cases are shown in Table 2 and Figure 3. The overall cut-out rate was 0.97%. The blade position between the CAB and NCAB groups did not differ significantly. An optimal blade position was achieved in 511 (82.4%) of the 620 cases. The implant positions are shown in Figure 2. No significant differences were found in the tip-apex distance between the two groups (CAB 16 ± 7.3 vs. NCAB 15.7 ± 7.6, *p* = 0.640) with an overall mean of 15.8 ± 7.5 mm.

### 3.3. Costs

The mean hospital reimbursement per case was USD 7154 ± 834. Implant costs for the nail system without augmentation (nail, blade, and locking screw) were USD 484 US dollars. For augmentation, the TRAUMACEM V+ Injectable Bone Cement system (Fa. DePuy Synthes) was used with an overall cost of USD 432. The price of the augmentation system included the TRAUMACEM™ V+ Injectable bone cement, TRAUMACEM™ V+ syringe kit and TRAUMACEM™ V+ Injection cannula (Fa. DePuy Synthes). The mean operating time was about 8 min longer in the CAB group (CAB 62.6 ± 21.2 min vs. NCAB 54.01 ± 7.1; *p* < 0.05).

### 3.4. Complications and Mortality

Surgical site infections occurred in only 0.97% of the cases without a significant difference between the groups. The post-operative hematoma was detected in 3.1% of patients, showing no significant difference between the groups. Hospital mortality (CAB 8.7% vs. NCAB 4.0%; *p* < 0.05; *n* = 620) and 1 year mortality (CAB 26.3% vs. NCAB 15.3%; *p* < 0.05; *n* = 313) were significantly higher in the cement augmentation group.

## 4. Discussion

Mechanical failure of osteosynthesis in the proximal femur is a feared complication when treating elderly patients with osteoporotic bone. In the literature, the cut-out of the femoral neck blade through the femoral head was reported in between 1.8% and 16.5% of cases [3,7,8,20]. In the study by Bojan et al. [9], a failure rate of 1.8% was observed in 3066 hip fractures treated with the gamma nail between 1990 and 2002. Davis et al. [5] investigated the causes of mechanical osteosynthesis failure in a prospective study of 230 hip screws and Y-nails. The study found 12.4% cut-out and 16.5% further implant failures.

The aim of this study was to evaluate the mechanical failure rate of a large number of patients using a cement-augmented helical femoral neck blade if patient characteristics suggested severe osteoporosis. Using cement augmentation of the blade in about 50% of the cases, the overall mechanical failure rate was lower than in the mentioned studies and evaluated under 1%. Notwithstanding, cement augmentation of the blade extended the operating time by 8 min and nearly doubled the implant costs. Overall, six patients showed signs of mechanical failure, resulting in revision surgery with arthroplasty. If revision surgery was required due to mechanical osteosynthesis failure, a high rate of complications, reoperation, significant loss of function, and pain can be expected [21]. 

After revision surgery, mortality increases up to 56% in the first year. In addition, according to Palmer et al. [10], a considerable increase in treatment costs with a simultaneous poor prognosis can also be expected. If cement augmentation proves to prevent costly revision surgery, the increased implant costs might be justified. To prevent mechanical failure, Erhart et al. [15] tested the anchorage of PFNA blades augmented with bone cement in eight fresh frozen femoral heads, showing increased rotational stability and higher pullout strength. There is limited clinical evidence evaluating cement augmentation of the femoral neck blade. 

Yee and Kammerlander et al. [16,17] addressed the cut-out rate of PFNA blades. Yee et al. [17] studied 76 patients (47 in the CAB group and 29 in the NCAB group) in their retrospective study. The cut-out rate was 2.1% in the CAB group and 13.8% in the NCAB group. Thirty-one patients (29%) could not be followed-up until 6 months postoperatively. In a prospective, multicenter, randomized, and patient-blinded study in 2018, Kammerlander et al. [16] investigated 223 patients (105 in the CAB group and 118 in the NCAB group). There was no cut-out in the CAB group compared to the six cases (5%) in the NCAB not reaching statistical significance due to the low number of cases. One conclusion of the study was that although blade augmentation does not improve patients’ ability to walk, it may have the potential to prevent revision surgery by strengthening the osteosynthesis construct. Compared with these studies, augmentation was performed in 48% of patients (299 of 620) in the present study. The average OR time with augmentation was 8 min longer, apart from the additional implant costs. The patients in the CAB group were, on average, 11 years older, and the proportion of women was significantly higher. Both values, age and female gender, are known risk factors for osteoporosis. Nevertheless, the cut-out rate between these two groups in our study showed no significant difference (CAB 1% vs. NCAB 0.9%). Therefore, despite the higher costs and not reaching statistical significance, this might confirm the higher stability of augmented femoral neck blades in patients with severe osteoporosis. In line with the literature [12,13,14,15], there was no significant difference in the cut-out rate between the groups. Due to the low overall cut-out rate of less than 1%, the study cohort needs to be even larger than the 620 evaluated patients.

With regard to the low cut-out rate in our NCAB group, the authors would like to emphasize the value of the anatomical reduction, tip-apex distance (TAD) of less than 25 mm, and a center-center or inferior-center blade position. In 2008, Lobo-Escolar et al. [3] showed in their case-control study that an unfavorable tip-apex distance and blade position are significant risk factors for cut-out. The study evaluated 916 proximal femur fractures treated with a gamma nail resulting in a mechanical failure rate of 3.3%. This is in line with several studies [22,23,24]. Ibrahim et al. [25] analyzed 313 patients for fracture pattern, TAD, Parker’s ratio and reduction quality and demonstrated that insufficient fracture reduction also was a significant predictor of implant failure.

The study has certain limitations. This is a retrospective study with a short follow-up of only 3 months in all patients and up to 1 year in only 313 patients. However, mechanical implant failure may occur even later. There is a huge selection bias due to the significant age and gender differences in both groups as well as the unrandomized design of the study. Furthermore, no functional or emotional data were collected. To show a significant difference between the groups, a large RCT with a long follow-up period must be performed because the cut-out rate of modern implants is very low with correct indications and surgical techniques. Clinical parameters, such as ROM (range of motion), HHS (Harris hip score), and VAS (visual analogue scale), etc., could then be included.

## 5. Conclusions

In conclusion, this study evaluated, by far, the largest cohort of patients with proximal femoral fractures treated with augmented blades compared to non-augmented blades. A significant difference between the groups could not be established despite increased implant costs and longer operating times. However, treating proximal femoral fractures with cephalomedullary nails can achieve a mechanical failure rate of less than 1% of anatomical reduction, a tip-apex distance of less than 25 mm, an optimal blade position is achieved, and cement augmentation is used in cases with severe osteoporosis. Therefore, the authors recommend the augmentation of the blade in osteoporotic patients. However, it should be noted that augmentation is expensive and prolongs surgery time without definite proof of its mechanical superiority.

## Figures and Tables

**Figure 1 jcm-12-01661-f001:**
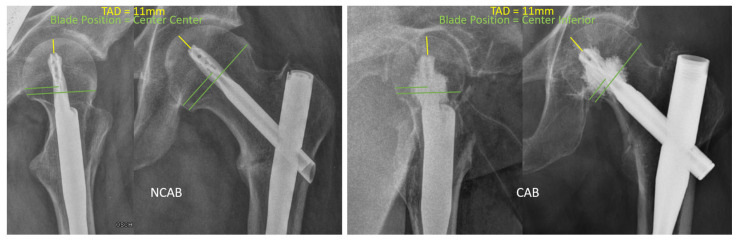
Measurement of the TAD (tip apex distance = yellow lines) and the blade position (position of the blade in the area of the femoral head = green lines). Left side: NCAB; right side: CAB.

**Figure 2 jcm-12-01661-f002:**
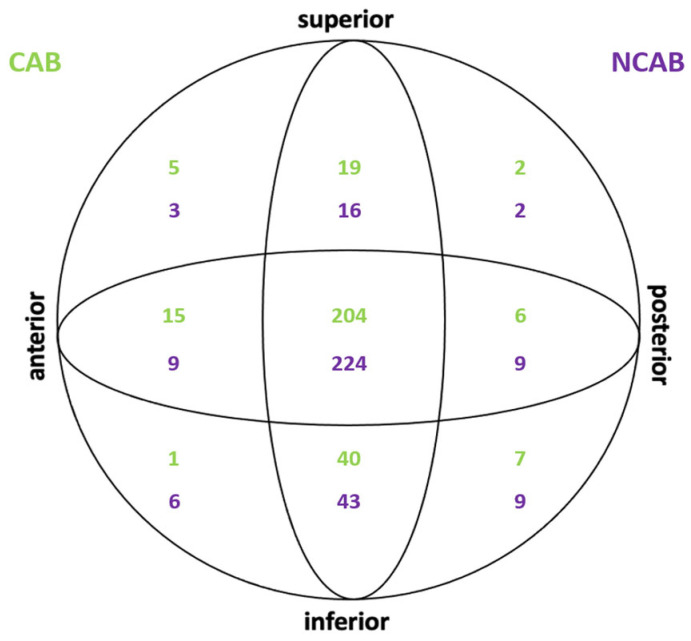
Blade positions of patients without mechanical failure.

**Figure 3 jcm-12-01661-f003:**
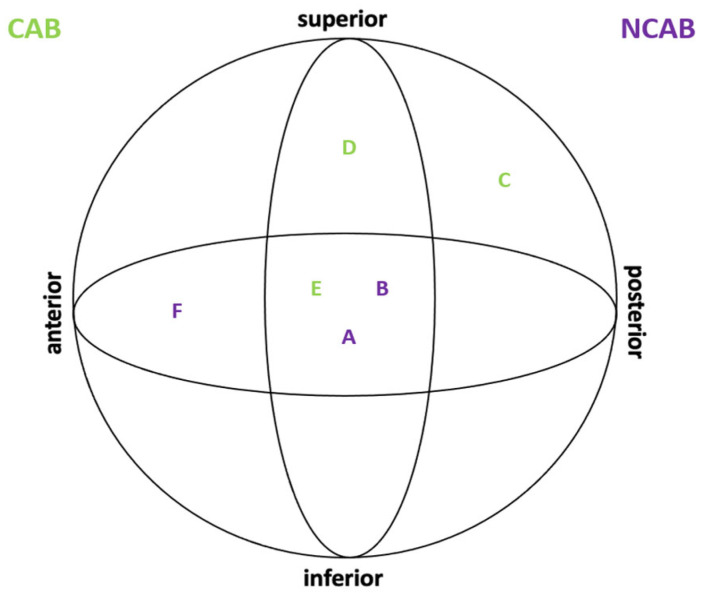
Blade positions of patients with mechanical failure.

**Table 1 jcm-12-01661-t001:** Patient demographics.

Variable	CAB	NCAB	*p*-Value (s/ns)
Patients	299	321	
Age (years)	86 ± 8	75 ± 15	***p* < 0.001 (Welch)**
Sex	m: 21.7% vs. f: 78.3%	m: 44.2% vs. f: 55.8%	***p* < 0.001 (Chi-Qua)**
AO classification			
31-A1	88	97	
31-A2	150	143	
31-A3	61	81	
ASA score			*p* = 0.13 (*t*-test)
1	2	9	
2	28	39	
3	199	205	
4	70	68	
Time to surgery [h]	10.6 ± 4.2	11.9 ± 14.8	*p* = 0.44 (Chi-Qua)
OP time [min]	62.6 ± 21.2	54.01 ± 7.1	***p* < 0.001(Chi-Qua)**
Hospital stay [days]	10.6 ± 4.2	11.9 ± 2.8	*p* = 0.66 (Chi-Qua)
Optimal blade position	81.6%	83.2%	*p* = 0.46 (Chi-Qua)
Tip-apex [mm]	16 ± 7.3	15,7 ± 7.6	*p* = 0.640 (T-Test)
Cut-out	3	3	*p* = 0.832 (Chi-Qua)
Hematoma	13	6	*p* = 0.101 (Fischer)
Surgical site infection	3	3	*p* = 1 (Fischer)
Hospital death	8.7%	4%	***p* < 0.017(Chi-Qua)**
One-year mortality	26.3%	15.3%	***p* < 0.016(Chi-Qua)**

The significant values have been highlighted in bold.

**Table 2 jcm-12-01661-t002:** All patients with mechanical failure.

Case	Age	Sex	AO-Classification	Tip Apex	First Signs	Augmentation
(A)	77	female	31-A3	16 mm	39 days	no
(B)	84	female	31-A2	19 mm	56 days	no
(C)	88	female	31-A2	16 mm	10 days	yes
(D)	98	female	31-A2	13 mm	14 days	yes
(E)	93	female	31-A3	2 mm	6 days	yes
(F)	65	female	31-A2	11 mm	41 days	no

## Data Availability

All authors decided that the data and material would not be deposited in a public repository.

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
