# Peer review of "Blade Augmentation in Nailing Proximal Femur Fractures—An Advantage despite Higher Costs?"

_jcm, 2023, doi:10.3390/jcm12041661_

Round 1

Reviewer 1 Report

Prevention of proximal femur fractures treatment complications and failures is the hot topic in locomotor system trauma specialists. This paper could be one of the significant voices in this discussion. The strongest side of the paper is the number of patient included to the study. Unfortunately, there are also several of weak points as in order:

1.      Type of the stud: the retrospective research

2.      The significant patients age diffrence in both groups

3.      Short follow-up

4.      The ASA score, just shortly named in section Patient population,  higher than 2 contains three categories of patients with different health burdens. It has little usefulnes for orthopeadid surgeons

5.      The text of results presentation is cursory and only Table 1 inclusion supports it

6.       The clear conclusions are absent

7.      The weak points of the paper named by authors are right but incomplete

Reviewer 2 Report

Thanks for the opportunity to review this manuscript. Some comments need to be revised by the author.

Materials and Methods: 

Line 69-71 What are the specific criteria for cement augmentation?

Line 75-85 There are few observation indicators in this study, so it is not easy to judge the surgical effectiveness between the two methods – e.g., Harris score, VAS score, and the range of motion. Authors should add observation indicators.

Line 88-92 The data analyzed by t-test conforms to the normal distribution. If the data does not conform to the normal distribution, non-parametric tests should be used.

Discussion: 

The paper comes across much too firmly on the side of CAB. Soften this paragraph.

The authors should describe that some of the results may relate to CAB v.s. NCAB – e.g., with the hip function.

Round 2

Reviewer 1 Report

Dear Authors,

thank you for changes which you done in this manuscript version. I'd like to recomend to accept paper in this version.
